# The Importance of Charge Transfer and Solvent Screening in the Interactions of Backbones and Functional Groups in Amino Acid Residues and Nucleotides

**DOI:** 10.3390/ijms232113514

**Published:** 2022-11-04

**Authors:** Vladimir Sladek, Dmitri G. Fedorov

**Affiliations:** 1Institute of Chemistry, Slovak Academy of Sciences, Dubravska cesta 9, 845 38 Bratislava, Slovakia; 2Research Center for Computational Design of Advanced Functional Materials (CD-FMat) National Institute of Advanced Industrial Science and Technology (AIST), Tsukuba 305-8568, Ibaraki, Japan

**Keywords:** fragment molecular orbital, molecular dynamics, molecular mechanics, non-covalent interactions, residue interaction networks, dielectric constant

## Abstract

Quantum mechanical (QM) calculations at the level of density-functional tight-binding are applied to a protein–DNA complex (PDB: 2o8b) consisting of 3763 atoms, averaging 100 snapshots from molecular dynamics simulations. A detailed comparison of QM and force field (Amber) results is presented. It is shown that, when solvent screening is taken into account, the contributions of the backbones are small, and the binding of nucleotides in the double helix is governed by the base–base interactions. On the other hand, the backbones can make a substantial contribution to the binding of amino acid residues to nucleotides and other residues. The effect of charge transfer on the interactions is also analyzed, revealing that the actual charge of nucleotides and amino acid residues can differ by as much as 6 and 8% from the formal integer charge, respectively. The effect of interactions on topological models (protein -residue networks) is elucidated.

## 1. Introduction

Proteins and nucleic acids are of paramount importance to life. Interactions between amino acid residues and nucleotides are key factors driving many processes in life sciences. Experimental evidence of the role of individual residues is typically indirect, based on mutations and differential properties of mutated species.

Computational studies can reveal contributions of residues to the binding in a given complex. Force fields have been very successful in describing molecular binding in biochemical systems [1]. However, they have their limitations, and it is desirable to compute interactions at a higher level using a quantum mechanical (QM) approach.

It is challenging to apply QM methods to biochemical systems because the cost of calculations scales steeply with their size. Regular QM methods provide total energies, but not interactions between parts of a system. There is a variety of low-scale fragment-based QM methods [2], which not only reduce the cost, but can also deliver the properties of fragments, including interactions between them.

The fragment molecular orbital (FMO) method [3,4,5] is one such approach. It has been applied to a variety of proteins and nucleic acids [6,7,8,9,10,11,12]. In this work, FMO is employed in combination with density-functional tight-binding (DFTB) [13] and the polarizable continuum model (PCM) [14].

Interaction energies [15,16] have been extensively used to analyze protein–ligand [17] and protein–protein binding, with the goal of designing better, more potent drugs in structure and interaction-based drug design (SIBDD) [18]. Interactions can be used as descriptors in structure–activity relationship (SAR) studies [19], for building models of protein residue networks (PRN) [20], and for coarse-grained force fields [21]. For a reliable analysis, interaction energies can be averaged for a selection of snapshots from MD simulations [9,22,23,24]. Interaction energies differ from binding energies in the treatment of polarization and desolvation [25].

An important goal of this work is to develop a methodology for analyzing interaction energies between any combination of amino acid residues and nucleotides, appearing in proteins, and their complexes with other proteins and DNA, including DNA aptamers. Aptamers are promising candidates for pharmaceutical applications [26,27].

Secondly, it is of interest to compare interactions in molecular mechanics (MM) and QM and discuss the role of polarization [28] and charge transfer. The polarization of proteins can be critical to ligand binding [29]. In MM, atomic charges are fixed, and the charges of residues are integer numbers. In reality, because of charge transfer, the actual charge of a residue should be fractional. It is a goal of this work to compute interactions for residues with fractional charges, obtained from QM calculations. Although there are force fields that are polarizable [30,31] and some with charge transfer models [32], most MM applications continue to use traditional non-polarizable force fields. It is important to discuss the role of solvent screening, especially for charged nucleotides and amino acid residues. In QM calculations, a robust definition of solvent screening can be obtained based on induced solvent charges [33].

Thirdly, many applications of the FMO rely on interactions between fragments, which are slightly different from conventional residues. In this work, interactions for both typical FMO fragments and conventional residues are computed on equal footing.

Finally, it is of interest to understand the role of backbones. Are they mere scaffoldings, or do they also participate in the binding? If yes, how much do they contribute? Their role has a direct implication for many pharmacological problems, because mutations affect the contributions of side chains, but not backbones (although a mutation can also cause a structural deformation). In particular, understanding the role of backbones can be helpful to rationalizing site-directed mutagenesis and alanine scanning [34].

Biochemical processes occur at physiological temperature and reflect the flexibility of macromolecules. Therefore, in this study, all interactions are averaged over a set of 100 snapshots obtained from molecular dynamics (MD) simulations of a protein–DNA complex.

Interactions between base pairs in nucleic acids have attracted considerable interest with a variety of theoretical [35,36,37,38,39,40] and experimental [41,42,43] results available. Interactions between residues in proteins and nucleotides have been analyzed with the FMO [27,44,45,46,47,48].

## 2. Methods

### 2.1. Modeling Protein–DNA Complex

The MutS enzyme (PDB ID: 2o8b [49]) was used as a case study for protein–DNA interactions. MutS detects errors induced in the DNA replication by DNA polymerases. The complex of a part of this enzyme with a DNA helix was studied by Bouchal et al. [50] using MD simulations, and this trajectory was used in this work to extract relevant snapshots for conducting QM calculations.

The truncated protein model contains residues from 362 to 532 of the full protein, whereas the DNA has 30 nucleic acid residues. There are 3763 atoms in total, of which 2814 and 949 atoms are in the protein and DNA, respectively. The forward sequence of the DNA double-helix is 5′-GAACCGCXCGCTAGG-3′ and the reverse sequence 5′-CCTAGCGYGCGGTTC-3′, with X = G, Y = T, i.e., a mismatch.

From the 50 ns MD trajectory [50], one snapshot per 10 ps was extracted (5000 snapshots in total). This set of snapshots was used to analyze structural changes in the trajectory. The RMSD of all heavy (non-hydrogen) atoms was computed relative to the first snapshot (Figure 1a). It can be seen that the structure stabilized after 20 ns, with minor RMSD oscillations between 3 and 4 Å. The RMSD histogram in Figure 1b indicates that the structure does not undergo any significant conformational change after the RMSD reaches the value of 3–4 Å, as the corresponding distribution of the RMSD is unimodal, indicating an absence of multiple distinct conformers. Therefore, for conducting QM calculations, the part of the MD trajectory between 20 and 50 ns was used, and 1 snapshot per 300 ps was extracted (100 snapshots in total).

### 2.2. Evaluation of Interactions

In the FMO method, fragments are defined in such a way to ensure the accuracy in the total properties such as energies or dipole moments. As a result, fragments may differ from the desirable units suitable for a discussion of their properties. FMO fragments in proteins differ from amino acid residues by a CO group, and nucleotides differ by a PO_3_ group, namely a residue fragment has the same atomic composition as a conventional residue, but each CO group comes from an adjacent residue, and likewise, phosphate groups are shifted in nucleotides (see Appendix A).

In the FMO, the pair interaction energy (PIE) $\Delta E_{IJ}$ between fragments quantifies the binding between two units (nucleotides, etc). In the pair interaction energy decomposition analysis (PIEDA) [23], PIEs are decomposed into components, whose nature depends on the QM method. For DFTB/PCM, a PIE is decomposed into electrostatic (ES), 0-body (electron exchange, correlation, and repulsion), charge transfer (CT) coupling to ES (CT·ES), dispersion (DI), and solvent screening (solv) terms.
(1)$$\Delta E_{IJ}=\Delta E_{IJ}^{\mathrm{ES}}+\Delta E_{IJ}^{0}+\Delta E_{IJ}^{\mathrm{CT}\cdot \mathrm{ES}}+\Delta E_{IJ}^{\mathrm{DI}}+\Delta E_{IJ}^{\mathrm{solv}}$$

One reason for choosing DFTB in this work is the availability of partition analysis (PA) [51], which has been applied to identify solvent molecules strongly bound to residues in protein crystals [52] and to clarify the transition state stabilization in solid state catalysis [53]. Conceptually, PA resembles partial methods [54].

In PA, the basic units for decomposing DFTB energies are called segments. Segments can be defined exactly as conventional residues, both for amino acids and for nucleotides. Technically, a segment definition is accomplished by using a PDB file, the same one in MM and QM, where atoms are assigned to residues (see Appendix A for details).

Fragments in the FMO are calculated with a QM method, so that there is an integer number of electrons and protons in each fragment, and hence, fragments themselves bear integer charges. This is similar to MM, where atomic charges in each residue sum up to an integer number. However, in the FMO, fragments are polarized by the electrostatic embedding, so that each fragment has a unique distribution of atomic charges. Charge transfer between fragments (residues), missing in MM, is accounted for at the two-body level in the FMO. On the other hand, in MM, each residue has a fixed set of atomic charges typically determined by atom types [55]. Carbon atoms with different atomic types (for example, Cα, and carboxyl C) have different atomic charges, albeit in each residue, a given atom type bears a fixed charge, unlike atoms in the FMO. Polarized protein-specific charges from QM can be used in MM simulations [29,56,57].

Segment charges in PA reflect both polarization and charge transfer, incorporated in segment energies (referred to as one-body terms), whereas interaction energies (two-body terms) have no explicit CT contribution and its coupling to the 0-order Hamiltonian, $\Delta E_{IJ}^{0}$.

To distinguish segments from fragments, fragments are numbered with capital letters such as *I* and *J*, whereas segments are numbered with small letters *i* and *j*. In PA, pair interaction energies are decomposed into
(2)$$\Delta E_{ij}=\Delta E_{ij}^{\mathrm{ES}}+\Delta E_{ij}^{\mathrm{DI}}+\Delta E_{ij}^{\mathrm{solv}}$$

To discuss solvent effects, the values of the PIEs for the solute are defined as
(3)$$\Delta E_{ij}^{\mathrm{solute}}=\Delta E_{ij}^{\mathrm{ES}}+\Delta E_{ij}^{\mathrm{DI}}$$

There is another difference between fragments and segments. Due to the treatment of covalent boundaries between fragments, a PIE for two fragments connected by a covalent bond is large (for DFTB, on the order of −300 kcal mol^−1^), but for segments, the values of such PIEs are on par with non-covalent interactions.

DFTB calculations were performed at the level of FMO2-DFTB3/C-PCM〈1〉 with D3(BJ) dispersion [58], 3ob [59] parameters, and partial screening [33], as implemented [60] in GAMESS [61].

In MM, pair interactions are decomposed into electrostatic (elec) and van der Waals (vdW) terms as
(4)$$\begin{array}{cc} \Delta E_{ij} & =\Delta E_{ij}^{\mathrm{elec}}+\Delta E_{ij}^{\mathrm{vdW}} \end{array}$$
(5)$$\begin{array}{cc} \Delta E_{ij}^{\mathrm{elec}} & =\sum_{A\in i}\sum_{B\in j}\frac{Q_{A}Q_{B}}{\varepsilon R_{AB}} \end{array}$$
(6)$$\begin{array}{cc} \Delta E_{ij}^{\mathrm{vdW}} & =\sum_{A\in i}\sum_{B\in j}\frac{C_{AB}^{12}}{R_{AB}^{12}}-\frac{C_{AB}^{6}}{R_{AB}^{6}} \end{array}$$
where *A* and *B* are the number of atoms, $R_{AB}$ is the distance between *A* and *B*, $Q_{A}$ is the charge of atom *A*, ε is the dielectric constant, and $C_{AB}^{12}$ and $C_{AB}^{6}$ are parameters. The AMBER force field imposes some restrictions on the sums (not explicitly shown in Equation (5)), excluding some close pairs of atoms A,B and weighing 1–4 interactions of terminal atoms of dihedrals [62].

In this work, the ff14SB [63] and Parmbsc1 [64] parameters were used for the protein and DNA, respectively, the same parameters as in the MD trajectory [50].

In Equation (5), ε is the relative permittivity of the medium, also known as the dielectric constant. It describes the solvent screening. In AMBER, it is conventional to use ε=1 [62]. Other values of ε are used in computing binding energies in Poisson–Boltzmann models [65]. The inability of force fields to take into account local variations of the electric field is often criticized as a drawback when evaluating accurate binding free energies [66,67]. On the other hand, in QM/PCM calculations, the locality of solvent screening is fully taken into account. Fragmentwise effective values of ε representing the local features of the solute/solvent interface can be defined in QM/PCM, although they are not directly used for computing the screening [68].

The actual dielectric constant of “dry” proteins was estimated to be ε=3.23±0.04 [69]; however, this global value does not describe local effects. The dielectric constant inside a protein may differ from that on its surface at the solute/solvent interface. It was assessed [70] that ε is 6…7 inside a protein and 20…30 on the surface. Elaborate schemes for dealing with solvent screening in MM have been proposed [71,72]; however, the bulk of common MD simulations continues to use the traditional vacuum form of the Coulomb interaction in the solute with ε=1, whereby the solute is in a mechanical embedding of the solvent, and both the solute and solvent are not polarizable.

In DFTB, the ES term is computed with a different functional form than in force fields, although ε=1 is likewise used, whereas the solvent screening is described in the separate solv term in Equation (2). In addition to the use of polarizable atomic charges in DFTB compared to fixed charges in MM, damping functions γ and Γ describe the effect of charge penetration for close interatomic contacts, as well as charge transfer via the chemical hardness in the third-order Γ terms [73].
(7)$$\Delta E_{ij}^{\mathrm{ES}}=\sum_{A\in i}\sum_{B\in j}\left\{\left(\gamma_{AB}+\frac{1}{3}\Gamma_{AB}Q_{A}+\frac{1}{3}\Gamma_{BA}Q_{B}\right)Q_{A}Q_{B}\right\}.$$

On the other hand, the dispersion in DFTB is conceptually very similar to the vdW term in force fields, a non-polarizable contribution, computed, however, with different parameters so that the values of dispersion in QM and MM differ.

### 2.3. Definition of Fragments and Segments

In this work, the structural units used for the analyses are as follows. In MM and PA, conventional nucleotides and amino acid residues were used (segments). FMO fragments have a shifted phosphate (nucleotides) or carbonyl (amino acids); see Appendix A.

Side chains in amino acid residues and bases in nucleotides are referred to as functional units (f). The remaining part in each unit is called the backbone (b). This splitting was only performed for segments in PA and MM, but not for fragments in the FMO.

A detailed atomic structure of all nucleotide pairs is shown in Figure 2 and Appendix A, where the atom names as used in the PDB are also indicated.

## 3. Results and Discussion

### 3.1. Interactions between Residues

Pair interaction energies for selected pairs (Appendix A), averaged over all 100 sampled structures, are shown in Table 1. For all interactions, the standard deviations are quite small, typically a few percent, indicating that the structure does not undergo major changes during the production part of the MD trajectory.

The most direct comparison of MM with QM can be performed for MM(solute,res) with QM(PA,solute,res), because MM interactions are between solute atoms without solvent screening, similar to the solute QM results. The correlation between MM and QM solute interactions is shown in Figure 3a. The correlation coefficient $R^{2}$ is high, 0.993. As can be seen from Table 1, solute pair interaction energies can be very large (up to −105.1 kcal mol^−1^), because they are computed in a vacuum (although QM solute results reflect the polarization of the solute by the solvent).

The last pair in Table 1, Met107:Asp108, is excluded from Figure 3a. This pair is special because there is a covalent bond between the two residues. The MM and QM interactions are −27.6 and −4.9 kcal mol^−1^, respectively. In MM, near-neighbor interactions between some atoms are excluded, but in QM, all pairs of atoms contribute. In QM(DFTB), the electrostatic interaction is not of a simple Coulomb form, but includes a damping in the form of the γ and Γ functions (Equation (7)) modeling charge penetration at short distances. To discern the origin of the QM and MM discrepancy, the ES term was calculated using MM charges for all atom pairs, which decreased the absolute value by only 2 kcal mol^−1^. Therefore, the bulk of the difference between −27.6 (MM) and −4.9 kcal mol^−1^ (QM) is due to other reasons, that is the damping, the three-body ES term in DFTB, and the distribution of atomic charges in the residues.

The largest PIEs are between charged residues (the formal charges of residues are shown in Appendix A). All nucleotides have the formal charges of −1. The atoms in the phosphate group bear most of that charge. Two negatively charged nucleotides may still have an overall attractive interaction, and the strong Coulomb repulsion of the phosphate groups is somewhat compensated by the attraction of the nucleoside bases.

The electrostatic interaction and solvent screening are large for ionic pairs, e.g., for the pair G10:Arg180, the value of −64.7 in QM(PA,solute,res) is reduced to −20.2 (kcal mol^−1^) in QM(PA,solution,res) by adding the solvent screening. A similarly large weakening of the interactions in solution is seen for the pair Lys86:Asp108, due to the addition of solvent screening $\Delta E_{ij}^{\mathrm{solv}}$. The inclusion of the screening can even change the sign of the PIEs, as seen for all A:T pairs and the mismatched G8:T23 pair. The slope in Figure 3a indicates that MM interactions are overestimated by a few % vs. QM.

Next, it is interesting to compare the interactions between conventional residues and shifted fragments. The results are shown in Figure 3b. Although there is a nice correlation for most pairs, there are two noticeable outliers, G10:Arg180 and C9:Arg180. The PO_3_ group is shifted in the FMO from C9 to G10, as shown in Appendix A. It can be seen in Appendix A that Arg180 interacts via its positively charged side chain with the negatively charged phosphate group of G10. The PO_3_ group of residue G10 is, however, assigned to fragment C9. The PIE of C9:Arg180 contains the interaction of NH${}_{2}^{+}$ from Arg180 and the negatively charged phosphate group from G10 and O3${}^{\prime }$ of C9. This is the reason for a mismatch between interactions as defined for fragments and segments in this case. If the pair interaction energies of C9:Arg180 and G10:Arg180 are added, then the res and frag results are very similar, −32.8 and −31.4 kcal mol^−1^, respectively. Thus, the difference between the results is due to the shifted definitions of units, fragments vs. residues.

A similar mismatch occurs in Gly78:Tyr96 and Trp82:Lys86. In the former, one H-bond between the phenolic OH group of Tyr96 and the CO group of the Gly78 backbone is assigned to fragment 79, not Gly78 fragment. For the latter pair, it is the CO backbone group of Trp82 that forms a H-bond with the NH backbone group of Lys86. The CO group of Trp82 is assigned to another fragment.

The fact that the point for the connected pair Met107:Asp108 does not stand out in Figure 3b (the regression line almost passes through it) can be seen as the success of PA in defining pair interaction energies for residues connected by a covalent bond.

Figure 3c shows the correlation between fragments in PA and the FMO. The point of this comparison is to probe the effect of charge transfer: in the FMO, fragments have integer charges reflecting polarization; in PA, charges are fractional, computed at a higher level. They reflect both polarization and charge transfer (the fractional and integer residue charges differ by about 2%). There is a high correlation between them with the correlation coefficient of $R^{2}$ = 0.991. The last pair Met107:Asp108 in Table 1 is excluded from this plot, also because of the difference of defining pair interactions related to a covalent bond. In the FMO, due to the boundary definition, the interaction energy is large, −394.0 kcal mol^−1^, but in PA, it is −5.9 kcal mol^−1^.

### 3.2. Charges of Residues

The charges of residues are shown in Table 2. The charges in the non-polarizable force field are fixed and do not depend on the conformation (no deviations). On the other hand, the charges in QM reflect conformation-specific polarization and charge transfer. It can be seen that the QM(res) from the MM(res) charges differs by at most about 8% (for Asp108), whereas for QM(frag), the deviations are slightly more, because of the difference in the definition of residues. In MM, charges are not specific, so that, for example, all A nucleotides have the same charge. In contrast, QM charges show variation depending on the environment.

The charges of functional units (side chains for amino acids and bases for DNA) in MM(res) and QM(res) are similar. The charge transfer amount of a few percent agrees with other studies of the many-body charge transfer [4,74].

It is interesting to observe that the division of the negative charge of −1 between the phosphate+pentose and base is roughly the same for all four types (the values for QM(res) are discussed next): the phosphate+pentose part obtains −0.75…−0.80, and the rest of the charge goes to the base. The bases in A, C, G, and T have the charges of −0.15, −0.14, −0.19, and −0.20, respectively. For amino acids, it is the side chain that bears almost all of the formal charge. Actual charges for side chains deviate within 9% from the total formal charges.

Other QM studies report a similar behavior. At the level of FMO2-MP2/6-31G*, natural population analysis yielded [46] the base charges of −0.17 for A, −0.18 for C, −0.16 for G, and −0.19 for T, which are in very good agreement with the DFTB Mulliken charges in this work.

### 3.3. Components of Interactions

For a more detailed comparison of QM and MM, individual components for pair interactions are shown in Table 3 and Figure 4 (the covalently bound pair of Met107:Asp108 is excluded). There is a good correlation of the electrostatic term, but no correlation for the dispersion. The solvent screening is an important contribution to the interaction energies.

### 3.4. Contributions of Backbones and Functional Units

To gain more insight, amino acid residues were split into backbones and side chains and nucleotides were split into the phosphate+pentose backbones and bases. For the uniformity of the notation, side chains and bases are called functional units. The contributions of backbones and functional units are shown in Table 4.

The most direct correspondence is between MM(res,vacuum) and QM(res,solute). MM calculations were performed in explicit solvent, which, however, did not affect pair interactions because MM is not polarizable and there is no solvent screening. On the other hand, QM(res,solute) calculations reflect solute polarization, to which solvent screening is added in QM(res,solution).

The correlation between the MM(res,vacuum) and QM(res, solute) results is shown in Figure 5. The correlation for backbone–backbone (bb) interactions is high, $R^{2}$ = 0.945, but the slope of 0.852 indicates that interactions in QM are systematically smaller than MM by about 15%. It is worth noting that bb interactions are relatively small, ranging between −7 and 15 kcal mol^−1^, but nevertheless, the interactions are strongly correlated. This is attributed to the electrostatic nature of these interactions.

For bf interactions, the correlation is even higher than for bb, with $R^{2}$ = 0.993, and a similar systematical difference of about 18% is observed between MM and QM. It may be noted that in the bf pairs, b is always negatively charged (as shown above, the backbone charge is about −0.8), whereas f may be both charged (cationic side chains in Arg180 and Lys86 and anionic side chain in Asp108) and neutral (the rest, although the side chains of nucleotides do have a charge of about −0.2).

On the other hand, fb interactions show low correlation when all points are considered, which is caused by a single outlier, Lys86:Asp108. The reason why Lys86:Asp108 is rather different in QM and MM may be that, in QM, there is a strong polarization of these residues, not described in MM.

For ff interactions, the correlation is high, $R^{2}$ = 0.994, and the slope indicates a systematic deviation of about 8% in the absolute values.

The effect of the solvent screening is included in the QM(res,solution) set of results in Table 4. These results reveal many interesting details about the interaction in solution.

First, backbone–backbone interactions for nucleotides are essentially zero (0.1 kcal mol^−1^). This is because nucleotides bind via the base and the anionic phosphate+pentose faces the solvent (or protein). Ignoring solvent screening would result in the opposite conclusion, that backbones are important and repel each other (10 kcal mol^−1^ at the QM level).

Second, backbone(amino acid)–backbone(nucleotide) interactions are likewise essentially zero (−0.1 kcal mol^−1^), for the same reason as for nucleotidic pairs. However, backbone–backbone interactions for two amino acid residues can be substantial. This is because peptide backbones have CO and NH groups that can form hydrogen bonds. These backbone–backbone interactions are essential for secondary structure stabilization and are not directly affected by mutations. It is very important for drug design to identify if an interaction is for the side chain or the functional group [34].

Third, both bf and fb interactions for nucleotides are totally screened (the largest is 0.2 kcal mol^−1^), attributed to the phosphates facing the solvent. As the backbone of amino acid residues is neutral, functional(nucleotide)–backbone(amino acid) interactions are also nearly zero (−0.1 kcal mol^−1^). However, for backbone(nucleotide)–functional(amino acid), there is some residual interaction for charged amino acids, which is only partially screened. Both bf and fb interactions for two amino acids may be substantial, because backbones can form hydrogen bonds.

Fourth, many (but not all) residue–residue interactions are dominated by the functional unit pairs (ff). For nucleotides, it is the only significant interaction. When one residue is an amino acid, then the functional unit in the amino acid is the principle contributor to the interaction as a rule, although for some pairs, the backbone can contribute as much or even more, as in the Trp82:Met107 pair.

### 3.5. Comparison of Base Pairs

In order to analyze the stability of natural and mismatched base pairs, the contributions of individual hydrogen bonds were computed by defining segments appropriate for the purpose. The details are shown in Figure 2 and Appendix A.

The results are shown in Table 5, where the total ff contribution in Table 4 is split among functional groups. Each functional unit (that is, a Watson–Crick base) is split into functional groups, as shown in Figure 2. For example, to obtain the interaction energy for HB1 in A13:T18, two small segments are defined, one for NH_2_ of A13 and another for CO of T18. There are many pairs of such small segments, and only the most important of them is explicitly listed in Table 5. The contribution of the rest of the segment pairs is summed up and listed as the “rest” term. The segment pairs with a substantial electrostatic interaction are listed as weak ES “bonds”.

As can be seen from Table 4, the three nucleotide pairs chosen for Table 5 are representative as there is little variation of interactions for a given pair of nucleotides due to the local environment.

To complement the energetics, bond distances were computed between the closest pair of heavy atoms in each bond. The main results are in Table 6, and the details of the statistical averaging are shown in Appendix A.

There is a clear correlation between $R_{AB}$ in Table 6 and the interaction $\Delta E_{ij}$ in Table 5: the shorter the bond, the stronger the interaction. Hydrogen bonds have a much shorter length of about 2.95 Å, compared to 3.53 Å for CO⋯H bonds [78].

### 3.6. Comparison of Energies

A comparison of computational and experimental results is summarized in Table 7. Due to variations in both the chemical composition (bases vs. nucleotides) and environment (artificially capped vs. naturally embedded), it is difficult to compare various values directly.

Many theoretical methods in a vacuum predict the binding between C and G bases to be between −25 and −28 kcal mol^−1^, both QM and MM, which is close to −22 kcal mol^−1^ for DFTB in this work and to one experimental enthalpy of −21.0 kcal mol^−1^.

For A:T bases, many theoretical estimates are between −12 and −15 kcal mol^−1^, whereas the DFTB result in this work is −9 kcal mol^−1^, and the experimental value is −13 kcal mol^−1^. For G:T bases, theoretical predictions are −10…−15, and the DFTB result in this work is −14 kcal mol^−1^. The general trend is that the binding decreases in the order of C:G, G:T, and A:T, consistent with the present work. Some part of the difference between DFTB and other QM methods may be due to the small basis set superposition error in DFTB, owing to not treating core electrons as particles, which leads to smaller values of interactions [23].

The MM interaction energies in this work have no explicit screening and the repulsion between the anionic basis overwhelms the attraction for A:T and G:T pairs, so they are repulsive. It is shown above that a similar trend was observed for the solute-only part of the interactions at the DFTB level.

### 3.7. Statistical Analysis of All Interactions in Topological Models

While it is useful to discuss individual interactions in detail, an integral measure can be defined for a complete set of all interactions. This can be conveniently performed with protein residue network (PRN) models.

These are topological representations of the interaction network in proteins (or other molecular systems) as graphs (Figure 6), where residues are represented as vertices (nodes) and the interactions between them as edges connecting the vertices [80,81,82,83]. According to the convention [20], only attractive interactions stronger than a threshold (usually −1 kcal mol^−1^) were used. The PRN can help to elucidate the role of individual residues [84] in static or dynamic models [85,86,87,88,89,90]. Macromolecular interfaces can be studied using singular-value decomposition (SVD) analysis [91] and network differential analysis (NDA) [92]. The mutual information (MI) [93,94] is useful to compare PRNs generated with different methods; see the Appendix A for details.

The differences in the PRN based on the FMO fragments and conventional residues are caused by the shift of the CO group in amino acids and the PO_3_ group in nucleotides; see Appendix A. In about 40% of cases (the coefficients of constraint defined in the Appendix A are $C^{\left(1|2\right)}\approx C^{\left(2|1\right)}\approx 0.4$), the two methods could predict the existence of an edge with a weight within ∼1 kcal mol^−1^ accuracy. Much of the measure quantifying the agreement comes from pairs for which $\Delta E_{ij}$ is small (between 0 and −1 kcal mol^−1^) without an edge in the graph (the region with small energies in Appendix A, the most populated). The nuc–nuc interactions are not much affected by the misassignment of the PO_3_ group as the dominant interacting groups are the nucleotide bases (Table 4), as indicated by the population of states near the diagonal in Appendix A. On the other hand, nuc–aa interactions in Appendix A are more affected by the shifted PO_3_, as these groups are orientated towards the protein. For the aa–aa interactions in Appendix A, the shift of the CO group can cause discrepancies in the PIE values if the backbone is involved in interactions. However, stronger interactions with $\Delta E_{ij}\leq -15$ kcal mol^−1^ usually come from charged side chains, and the CO shift does not affect them much (states closer to the diagonal in Appendix A).

To investigate the effect of fractional versus integer charges of interacting units, the FMO/solution/frag and PA/solution/frag interactions were compared, for which the atomic constitution is identical. The mutual information (MI) is larger, and the coefficients of constraint exceed 0.66. Appendix A shows that the joint probability distribution is more diagonal in all residue type combinations. That is, off-diagonal elements are less frequent, indicating that the mutual prediction of equivalent edge weight is to be expected. The change of the treatment of charge transfer in the monomer (PA) vs. dimer (FMO) terms can shift the PIEs by a few kcal mol^−1^ only.

The final comparison is for PA/solution/res vs. MM/solute/res, for which atomic compositions are identical and the focus is on comparing QM and MM. In MM, pair interactions are overestimated (no screening), and thus, there are 4–5-times more edges in the graph (an edge is added if $E_{\lim}$ is stronger than −1 kcal mol^−1^); see Appendix A. Two methods, using integer charges while neglecting the screening, FMO/solution/frag and MM/solute/res can be compared (Table 8). The coefficients of constraint are virtually unchanged for the nuc–nuc interactions. As discussed above, the effect of the shifted PO_3_ assignment is negligible in nuc–nuc interactions. Hence, the difference in the PIEs between PA (or FMO) vs. MM is mainly due to the inclusion of solvent screening.

Because MM overestimates the interaction due to the lack of solvent screening (in the conventional AMBER model), one can ask whether MM-based PRNs are a reliable representation of residue interaction networks. Their choice as possible features in advanced analysis of MD trajectories remains to be tested [96]. Especially alarming is that some residue pairs (A:T) have a repulsive interaction in MM, about +8 kcal mol^−1^, in contrast to about −8 kcal mol^−1^ in PA; see Table 1. Similarly, C:G interactions are −8 kcal mol^−1^ in MM against −21 kcal mol^−1^ in PA. The lack of screening may be a problem for nucleotide interactions beyond those of pairs in a double-helix [97], as occur in packed DNA, in non-canonical or transitional structures [98,99,100,101], or in complexes with ligands [102,103,104].

## 4. Conclusions

A methodology for analyzing interactions in proteins, DNA, and their complexes was developed, based on QM and MM calculations, decomposed into appropriate units: backbones, bases, and side chains. This methodology is facilitated by the convenient framework offered by the partition analysis developed for the fragment molecular orbital method combined with density-functional tight-binding.

As a case study of applying this methodology, a protein–DNA complex was computed and analyzed in detail. For a realistic treatment of temperature, 100 snapshots from MD simulations were averaged.

It was shown that there is good agreement between the MM and solute QM results, when both methods do not include solvent screening. Both a very high correlation and agreement in absolute values were found for the studied variety of interactions between nucleotides and amino acid residues. However, this agreement was observed only for the electrostatic interaction, but not for dispersion.

Nevertheless, some deviations were observed, as much as 5 kcal mol^−1^. These deviations were attributed to charge transfer facilitated by hydrogen bonds, which was as large as 6 and 8% of the formal charge for nucleotides and amino acid residues, respectively,

The addition of solvent screening in the DFTB/PCM calculations resulted in a pronounced change of the picture of the interactions. The repulsive A:T interactions (between anionic nucleotides) became attractive when the screening contribution was added. One can question if MD simulations of DNA with a non-polarizable force field without a solvent screening are reliable at all.

A separate point of interest is the comparison of conventional residues to standard fragments in the FMO, which differ in the backbone definition. It was found that, for most pairs, where the interaction was between base pairs or between a base and a side chain, there was very good agreement, but for some adjacent nucleotide–residue pairs (as G10:Arg180 and C9:Arg180), the difference in the backbone definition of amino acid residues resulted in a very different definition of the interactions, which may be a source of confusion in discussing the results. This misassignment affects particular interactions, but not their total sum.

To provide a composite measure of the difference in two sets of interactions, a statistical analysis of all attractive residue–residue interactions was conducted using PRN models. The inclusion of charge transfer (PA vs. FMO) had on average a small effect on the PIEs, and the two sets had a high correlation. The use of non-conventional residues (FMO fragments) can result in larger differences, as much as 20 kcal mol^−1^ in some pairs, attributed to the shift in the unit definition. QM and MM have large discrepancies because solvent screening is not accounted for in MM.

The definition of segments in PA makes it possible to examine the strength of individual hydrogen bonds between nucleotide bases. A:T pairs have two conventional H-bonds; C:G pairs have three. The mismatched G:T pair has two H-bonds, similar to A:T in strength, albeit its additional CO⋯H and ES1 interactions place G:T between A:T and C:G pairs, in agreement with other reports.

The interaction energies obtained in this work were compared to the binding energies reported in other studies, and in general, good agreement was observed. The developed methodology can be used in future applications of the FMO to biochemical studies such as drug design, where its ability to quantify the contributions of individual functional groups can be very useful.

## Figures and Tables

**Figure 1 ijms-23-13514-f001:**
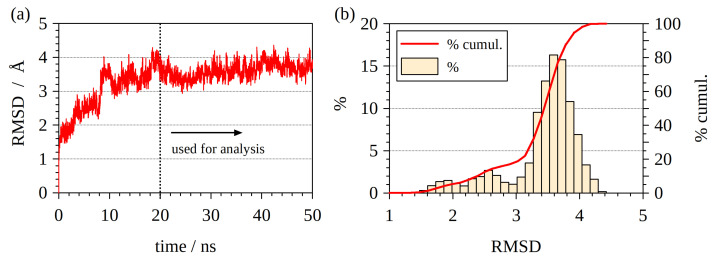
(**a**) RMSD (root mean square displacement) of heavy atoms in the trajectory with respect to the first snapshot. (**b**) Histogram of RMSD values. The distribution has a single major peak.

**Figure 2 ijms-23-13514-f002:**
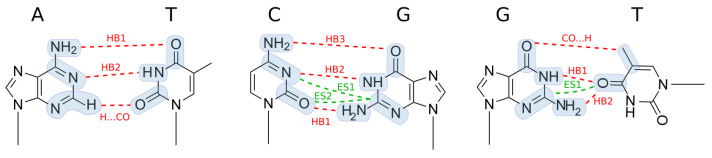
Three types of base pairs, adenine–thymine (A13:T18, regular), cytosine–guanine (C7:G24, regular), and guanine–thymine (G8:T23, mismatch). Hydrogen bonds (HB) and electrostatic (ES) interactions between functional groups (shadowed in grey) are labeled. See also Appendix A for more details.

**Figure 3 ijms-23-13514-f003:**
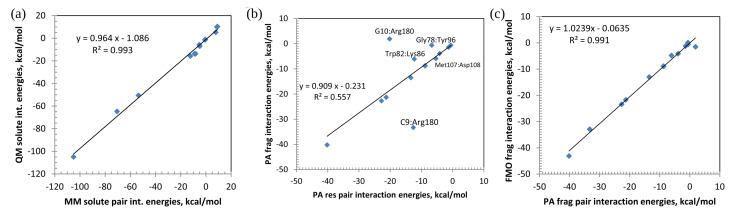
Correlation between pair interactions for conventional residues (res) and fragments (frag): (**a**) MM(solute,res) vs. QM(PA,solute,res), (**b**) PA(solution,res) vs. PA(solution,frag), and (**c**) PA(solution,frag) vs. FMO(solution,frag).

**Figure 4 ijms-23-13514-f004:**
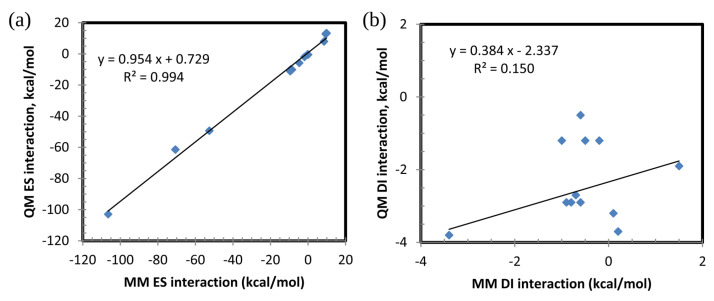
Correlation between components of pair interactions for conventional residues (res), MM vs. QM: (**a**) electrostatics (ES vs. elec) and (**b**) dispersion (DI vs. vdW).

**Figure 5 ijms-23-13514-f005:**
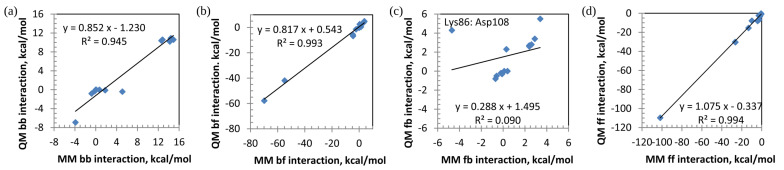
Correlation between pair interactions for conventional residues (res) divided into backbone (b) and functional (f) parts, MM vs. QM (solute): (**a**) bb, (**b**) bf, (**c**) fb, and (**d**) ff. For the distinction between bf and fb, see Table 4.

**Figure 6 ijms-23-13514-f006:**
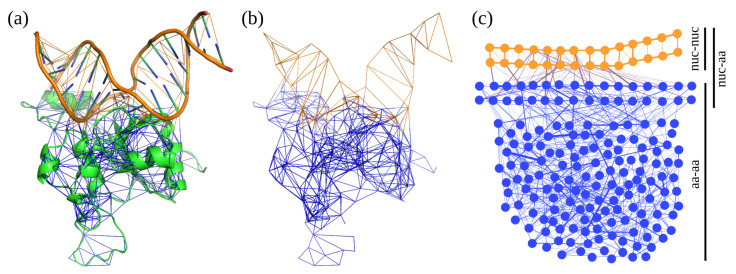
PRN models of the protein–DNA complex: (**a**) the PRN representation overlaid with the structure; (**b**) only the PRN without the structure; (**c**) a flat layout with the line thickness proportional to the interaction energy. Parts a-b are depicted with pyProGA [92], and part c is drawn with Gephi [95]. Orange and blue color denotes nucleotides (nuc) and amino acid (aa) residues, respectively.

**Table 1 ijms-23-13514-t001:** Selected pair interaction energies (kcal mol^−1^) in the protein–DNA complex (PDB: 2o8b), averaged over 100 snapshots of the MD trajectory ^*a*^.

Pair	MM(Amber)	QM(PA)	QM(PA)	QM(PA)	QM(FMO)
(State,Unit)	Solute,Res	Solute,Res	Solution,Res	Solution,Frag	Solution,Frag
(Unit Charge)	Integer	Fractional	Fractional	Fractional	Integer
(Definition)	$\Delta E_{\mathrm{ij}}$ in Equation (Equation 4)	$\Delta E_{\mathrm{ij}}$ in Equation (Equation 3)	$\Delta E_{\mathrm{ij}}$ in Equation (Equation 2)	$\Delta E_{\mathrm{ij}}$ in Equation (Equation 2)	$\Delta E_{\mathrm{IJ}}$ in Equation (Equation 1)
C7:G24	−8.3±1.8	−13.7±2.7	−21.3±2.3	−21.3±2.3	−21.7±2.1
A13:T18	8.8±1.4	10.1±1.8	−8.8±1.4	−8.8±1.4	−8.9±1.1
G6:C25	−9.0±1.6	−13.7±2.8	−22.8±2.1	−22.7±2.1	−23.4±1.6
A3:T28	8.8±1.3	9.9±1.9	−8.8±1.2	−8.7±1.2	−8.9±1.2
A2:T29	9.1±1.3	10.4±1.7	−8.9±1.2	−8.8±1.2	−8.9±1.1
G8:T23	7.8±1.2	5.3±1.8	−13.4±1.6	−13.5±1.6	−13.0±1.5
G10:Arg180	−70.6±2.6	−64.7±4.0	−20.2±3.0	1.9±0.5	−1.5±0.5
C9:Arg180	−53.6±2.7	−50.5±3.2	−12.6±2.0	−33.3±2.9	−32.9±2.8
G24:Phe101	−5.3±0.6	−5.8±1.1	−4.2±0.6	−3.9±0.6	−4.1±0.5
C7:Phe101	−0.6±0.4	−1.0±1.1	−0.6±0.3	−0.6±0.3	0.1±0.5
Gly78:Tyr96	−4.9±1.6	−7.0±1.7	−6.7±1.8	−0.6±0.2	−0.6±0.3
Trp82:Lys86	−12.4±1.2	−15.7±2.8	−12.3±1.5	−6.1±0.9	−4.8±1.0
Trp82:Met107	−0.7±0.6	−1.4±0.3	−1.3±0.3	−1.4±0.3	−1.2±0.2
Lys86:Asp108	−105.1±3.2	−104.9±4.0	−40.2±3.7	−40.2±3.7	−43.1±3.9
Met107:Asp108	−27.6±1.6	−4.9±2.3	−5.4±0.8	−5.9±0.9	−394.0±14.5

^*a*^ QM at the level of FMO2-DFTB3/PCM/D3(BJ)/3ob; res refers to conventional residues, whereas frag denotes conventional fragments in the FMO. Integer or fractional refer to the total charge on residues. The interaction energies in solution include solvent screening

**Table 2 ijms-23-13514-t002:** Charges (a.u.) of residues averaged in MD ^*a*^.

Residue	MM(total) ^*b*^	QM(res,total)	QM(frag,total)	MM(funct) ^*b*^	QM(res,funct)
A2	−1.0	−0.990±0.035	−0.988±0.023	−0.105	−0.155±0.017
A3	−1.0	−0.986±0.033	−0.987±0.022	−0.105	−0.153±0.016
G6	−1.0	−1.025±0.034	−1.024±0.021	−0.089	−0.195±0.018
C7	−1.0	−0.967±0.036	−0.956±0.027	−0.063	−0.141±0.018
G8	−1.0	−1.012±0.034	−1.014±0.022	−0.089	−0.191±0.016
C9	−1.0	−0.957±0.035	−0.918±0.024	−0.063	−0.148±0.016
G10	−1.0	−0.955±0.033	−0.970±0.029	−0.089	−0.189±0.017
A13	−1.0	−0.990±0.033	−0.983±0.023	−0.105	−0.154±0.016
T18	−1.0	−1.018±0.027	−1.018±0.018	−0.127	−0.205±0.014
T23	−1.0	−0.986±0.032	−0.985±0.022	−0.127	−0.193±0.016
G24	−1.0	−0.994±0.037	−0.967±0.028	−0.089	−0.195±0.020
C25	−1.0	−0.941±0.030	−0.964±0.022	−0.063	−0.144±0.018
T28	−1.0	−1.012±0.034	−1.015±0.021	−0.127	−0.204±0.017
T29	−1.0	−1.017±0.033	−1.009±0.022	−0.127	−0.205±0.017
Gly78 ^*c*^	0.0	−0.014±0.038	−0.015±0.019		
Trp82	0.0	0.016±0.040	0.009±0.018	0.029	0.038±0.010
Lys86	1.0	0.956±0.037	0.929±0.027	1.026	0.981±0.018
Tyr96	0.0	−0.009±0.037	−0.020±0.022	0.028	0.010±0.013
Phe101	0.0	−0.003±0.033	−0.012±0.019	0.019	0.014±0.009
Met107	0.0	−0.003±0.034	−0.007±0.023	0.050	0.041±0.010
Asp108	−1.0	−0.921±0.035	−0.917±0.022	−0.858	−0.911±0.015
Arg180	1.0	0.924±0.043	0.914±0.025	1.036	0.943±0.015

^*a*^ QM at the level of PA combined with FMO2-DFTB3/PCM/D3(BJ)/3ob. Total refers to the total charge; funct is the charge of the functional part (side chain for amino acids and base for DNA). Here, res refers to conventional residues, whereas frag denotes fragments, which are shifted by a few atoms. ^*b*^ All residues of the same type in MM have the same charges (computed from the .prmtop file), identical to published RESP values [55]. ^*c*^ Glycines have no side chains.

**Table 3 ijms-23-13514-t003:** Components of pair interaction energies (kcal mol^−1^) in the protein–DNA complex, averaged over 100 snapshots of the MD trajectory.

	MM(res) ^*a*^		QM(PA,res) ^*b*^		
Pair	$\Delta E_{\mathrm{ij}}^{\mathrm{elec}}$	$\Delta E_{\mathrm{ij}}^{\mathrm{vdW}}$	$\Delta E_{\mathrm{ij}}^{\mathrm{ES}}$	$\Delta E_{\mathrm{ij}}^{\mathrm{DI}}$	$\Delta E_{\mathrm{ij}}^{\mathrm{solv}}$
C7:G24	−8.6±2.8	0.2±1.9	−10.1±2.7	−3.7±0.2	−7.6±1.7
A13:T18	9.5±1.9	−0.8±1.2	13.1±1.7	−2.9±0.3	−18.9±1.2
G6:C25	−9.3±2.8	0.2±1.8	−10.0±2.8	−3.7±0.2	−9.0±1.5
A3:T28	9.4±1.7	−0.6±1.2	12.8±1.8	−2.9±0.2	−18.7±1.4
A2:T29	9.9±1.9	−0.9±1.3	13.3±1.7	−2.9±0.3	−19.3±1.4
G8:T23	8.5±1.5	−0.7±1.1	8.0±1.8	−2.7±0.3	−18.7±1.5
G10:Arg180	−70.7±3.2	0.1±1.3	−61.4±3.9	−3.2±0.3	44.4±2.3
C9:Arg180	−52.6±2.7	−1.0±0.3	−49.3±3.2	−1.2±0.2	37.9±2.2
G24:Phe101	−1.9±0.3	−3.4±0.5	−1.9±0.9	−3.8±0.5	1.6±0.9
C7:Phe101	0.0±0.3	−0.6±0.3	−0.5±1.0	−0.5±0.2	0.4±1.0
Gly78:Tyr96	−4.7±1.9	−0.2±0.9	−5.8±1.7	−1.2±0.2	0.2±0.5
Trp82:Lys86	−9.5±1.2	−2.9±0.8	−11.1±2.7	−4.6±0.6	3.4±2.0
Trp82:Met107	−0.2±0.1	−0.5±0.6	−0.2±0.1	−1.2±0.2	0.1±0.1
Lys86:Asp108	−106.5±4.4	1.5±1.7	−102.9±3.9	−1.9±0.2	64.6±3.3
Met107:Asp108	−26.8±1.5	−0.8±0.8	−0.4±2.3	−4.5±0.3	−0.5±1.8

^*a*^ The sum of MM terms (Equation (4)) is given in Table 1 as MM(Amber). ^*b*^ The sum of QM terms (Equation (2)) is given in Table 1 as QM(PA,total,res), whereas the sum (Equation (3)) is QM(PA,solute,res). QM calculations are at the FMO2-DFTB3/PCM/D3(BJ)/3ob level.

**Table 4 ijms-23-13514-t004:** Contributions of backbones (b) and functional units (f) to the total interaction energies (kcal mol^−1^) ^*a*^, averaged in MD.

	MM(res,vacuum)				QM(res,solute)				QM(res,solution)			
Pair	bb	bf	fb	ff	bb	bf	fb	ff	bb	bf	fb	ff
C7:G24	14.9	0.4	2.9	−26.4	10.6	2.6	3.4	−30.4	0.1	0.2	0.0	−21.7
A13:T18	12.6	3.8	2.4	−9.9	10.4	4.9	2.7	−7.9	0.1	0.0	−0.2	−8.8
G6:C25	14.2	3.2	0.3	−26.7	10.2	3.9	2.3	−30.2	0.1	−0.1	0.2	−23.0
A3:T28	12.6	3.8	2.4	−10.0	10.4	4.8	2.6	−7.9	0.1	0.0	−0.2	−8.8
A2:T29	12.8	3.8	2.5	−10.0	10.6	4.9	2.8	−7.9	0.1	0.0	−0.2	−8.9
G8:T23	14.4	3.3	3.4	−13.3	10.9	4.1	5.5	−15.4	0.1	0.1	0.1	−13.8
G10:Arg180	1.8	−69.9	0.1	−2.6	−0.1	−57.8	−0.0	−6.8	−0.1	−20.1	0.0	−0.1
C9:Arg180	5.1	−54.9	0.4	−4.2	−0.4	−41.9	−0.0	−8.1	0.0	−11.8	0.0	−0.8
G24:Phe101	−0.8	−2.3	−0.2	−1.9	−0.8	−1.4	−0.2	−3.3	−0.1	−1.0	−0.1	−2.9
C7:Phe101	0.7	0.0	−0.6	−0.7	0.0	−0.2	−0.5	−0.3	0.0	−0.1	−0.1	−0.4
Gly78:Tyr96	−0.2	−4.5			−0.3	−6.7			0.0	−6.7		
Trp82:Lys86	−3.9	−4.8	−0.7	−3.0	−6.9	−5.5	−0.8	−2.4	−6.2	−1.5	−0.6	−3.9
Trp82:Met107	0.0	0.0	−0.1	−0.6	0.0	0.0	−0.3	−1.1	0.0	0.0	−0.2	−1.1
Lys86:Asp108	0.1	1.3	−4.7	−101.8	0.0	0.5	4.3	−109.7	0.0	0.0	1.3	−41.5
Met107:Asp108	−23.0	−1.6	−1.5	−1.5	−0.9	−0.1	−1.6	−2.2	−1.7	−1.8	−1.7	−0.1

^*a*^ Standard deviations are not shown. As an example to explain the notation, bf for C7:G24 means the backbone (b) of C7 and the functional (f) unit of G24. QM calculations are at the level of PA with FMO2-DFTB3/PCM/D3(BJ)/3ob. Glycines have no side chains.

**Table 5 ijms-23-13514-t005:** Interactions $\Delta E_{ij}$ (kcal mol^−1^) between functional groups forming hydrogen bonds (HBs), CO⋯H bonds, and electrostatic (ES) bonds in selected base pairs in the protein–DNA complex, averaged over 100 snapshots of the MD trajectory ${}^{a}$.

Bond	A13:T18	C7:G24	G8:T23
HB1	−3.7±1.2	−4.8±1.1	−3.5±1.1
HB2	−4.3±0.6	−4.4±0.6	−4.3±0.9
HB3		−4.9±0.9	
CO⋯H	−0.8±0.4		−1.1±0.4
ES1		−2.2±0.4	−2.7±0.4
ES2		−1.0±0.5	
rest	0.0±0.5	−4.4±1.2	−2.2±0.9
total	−8.8±1.4	−21.7±2.3	−13.8±1.6

^*a*^ Computed at the QM(res,solution) level with FMO2-DFTB3/PCM/D3(BJ)/3ob. The total value here is equal to the ff value in Table 4. See Figure 2 for a picture of the bonds.

**Table 6 ijms-23-13514-t006:** Theoretical and experimental bond lengths $R_{AB}$ (Å) in Watson–Crick base pairs ^*a*^.

Pair	Bond	*A*	*B*	Amber ^*b*^	MP2/TRVPP ^*c*^	BP86/TZ2P ^*d*^	Expt. ^*e*^
A13:T18	HB1	N6	O4	3.04±0.22	2.86	2.85	2.95/2.93
	HB2	N1	N3	2.95±0.11	2.83	2.81	2.82/2.85
	CO⋯H	C2	O2	3.59±0.26			
C7:G24	HB1	N1	N3	2.96±0.10	2.90	2.88	2.95
	HB2	O6	N4	2.90±0.13	2.75	2.73	2.91
	HB3	O2	N2	2.88±0.13	2.89	2.87	2.86
G8:T23	HB1	O4	N1	3.03±0.18			
	HB2	O4	N2	2.87±0.11			
	CO⋯H	C7	O6	3.47±0.29			

^*a*^ Atoms A ∈ first base and B ∈ second base are the atom names used in the PDB file. ^*b*^ Base pairs are embedded in the solvated protein–DNA complex, from the MD trajectory [50] processed in this work. ^c^ RI-MP2/TRVPP geometries optimized in a vacuum [35]. ^*d*^ BP86/TZ2P geometries optimized in a vacuum [75]. ^*e*^ X-ray crystallographic measurements in sodium adenylyl-3′,5′-uridine hexahydrate [76] as a model of a A:T pair (two values for different environments) and in sodium guanylyl-3′,5′-cytidine nonahydrate [77] as a model of a C:G pair.

**Table 7 ijms-23-13514-t007:** Association energies ^*a*^ (kcal mol^−1^) for neutral Watson–Crick base pairs and anionic nucleotides.

Embedding	Pair	Solvent	Method	Reference	C:G	A:T	G:T
none ^*b*^	nucleotides	vacuum ^*b*^	AMBER	this work	−8.7	8.9	7.8
embedded ^*c*^	nucleotides	water	M06-2X/6-31G(d,p)/PCM	[38]	−16.8	−10.1	
embedded	nucleotides	water	FMO2-DFTB3/PCM	this work	−22	−8.8	−13.4
none	bases	vacuum	AMBER (parameters of Ref. [55])	[36]	−28.0	−12.8	
none ^*b*^	bases	vacuum ^*b*^	AMBER	this work	−26.6	−10.0	−13.3
none	bases	vacuum	CNDO-CI	[41]	−16.79	−7.00	−9.86
none	bases	vacuum	B3LYP/6-31G **	[35]	−25.5	−12.3	
none	bases	vacuum	B3LYP/D95 **	[37]	−26.65	−12.56	
none	bases	vacuum	RI-MP2/a[TQ]Z ^*d*^	[35,39]	−28.2	−15.4	
none	bases	vacuum	MP2/D95 **	[37]	−24.50	−11.69	
none	bases	vacuum	IQA/M06-2X ^*e*^	[40]	−29.1	−14.4	−14.6
embedded	bases	water ^*f*^	FMO2-MP2/6-31G *	[46]	−51 ^*g*^	−20	
embedded	bases	vacuum	FMO2-MP3/CP/MCP/dzp	[47]	−38	−13	
embedded	bases	vacuum	FMO2-CCSD(T)/MCP/dzp	[47]	−45	−20	
embedded	bases	water	FMO2-DFTB3/PCM	this work	−22	−8.8	−13.8
none	bases ^*h*^	none	Experiment(enthalpy)	[42]	−21.0	−13.0	
none	bases	none	Experiment(enthalpy)	[43]	−10.75		

^*a*^ Interaction energies for FMO methods; otherwise, binding energies. Pairs are hydrogen capped in standalone model systems (no embedding) or computed inside actual DNA (embedded). Watson–Crick bases are conceptually comparable to the ff values in Table 4. ^*b*^ In mechanical embedding. ^*c*^ In a A-DNA trimer duplex. ^*d*^ aTZ → aQZ BSL extrapolated [79]. ^*e*^ Bond dissociation energies. ^*f*^ With solvent polarization, but no explicit screening. ^*g*^ There is some variation for different pairs, and the average is given with the number of significant figures determined by the variation. ^*h*^ Methylated monomers, measured in vapor.

**Table 8 ijms-23-13514-t008:** Mutual information (MI) and coefficients of constraint *C* between two different interaction graphs ($\Delta E_{ij}$), labeled by 1 and 2 ^*a*^.

Graph 1	Graph 2		All	nuc–nuc	nuc–aa	aa–aa
FMO solution frag	PA solution res	MI${}^{\left(1;2\right)}$	0.1004	0.5269	0.0754	0.1040
		$C^{\left(1|2\right)}$	0.4009	0.9268	0.5794	0.3728
		$C^{\left(2|1\right)}$	0.4207	0.9432	0.6592	0.3872
FMO solution frag	PA solution frag	MI${}^{\left(1;2\right)}$	0.1647	0.5249	0.0868	0.1813
		$C^{\left(1|2\right)}$	0.6666	0.9271	0.7873	0.6491
		$C^{\left(2|1\right)}$	0.6899	0.9397	0.7592	0.6748
PA solution res	MM solute res	MI${}^{\left(1;2\right)}$	0.0684	0.1669	0.0604	0.0859
		$C^{\left(1|2\right)}$	0.0868	0.2233	0.0500	0.1315
		$C^{\left(2|1\right)}$	0.2732	0.2936	0.4643	0.3079
FMO solution frag	MM solute res	MI${}^{\left(1;2\right)}$	0.0459	0.1672	0.0486	0.0548
		$C^{\left(1|2\right)}$	0.0582	0.2236	0.0402	0.0838
		$C^{\left(2|1\right)}$	0.1922	0.2993	0.4249	0.2039

^*a*^ Averaged over 100 structures, using 40 bins in the histogram analysis with the parameter *E*_lim_ = −1 kcal mol^−1^. Graphs are labeled by the computational method, as in Table 1. The values are computed either for all pairs or limited to pairs shown (nuc and aa stand for nucleotide and amino acid residues, respectively).

## Data Availability

Raw data (100 structures used for averaging and FMO output files), as well as the software used for the analysis of the data are available on GitLab: https://gitlab.com/Vlado_S/charge-transfer-and-solvent-screening-in-the-interactions-of-residues-in-biomolecules (accessed on 14 October 2022).

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
