# Peer review of "The Importance of Charge Transfer and Solvent Screening in the Interactions of Backbones and Functional Groups in Amino Acid Residues and Nucleotides"

_ijms, 2022, doi:10.3390/ijms232113514_

Round 1

Reviewer 1 Report

In this manuscript, based on QM (DFTB) and MM calculations, a partition scheme was proposed for analyzing interactions in proteins, DNA and their complexes. The author declared that this scheme can be used for protein-DNA interactions, in which the role of charge transfer and solvent screening can be explored.

In general, this manuscript is not easily to read. I met lots of difficulties in the attempt for understanding. It is hope that the readability and reliability of this manuscript should be improved.

1.     Many important statements are collected in supporting information (Figure S1). Why not move it to the main-text? Additionally, it is better that provide a computational procedure for the DFTB based partition scheme.

2.     In 100th row, the definition for segment, “the basic units for decomposing DFTB energies are called segments” , is not clear. I can not find the difference between segment and fragment in Figure S1. Can they be expressed mathematically?

3.     In 108th row, the sentence “Conceptually, PA resembles partial methods.” is not clear.

4.     What’s the relationship between the interaction terms in eq. (1) and eq. (2)? Is DEIJ in eq. 1 equal to DEij in eq. 2?

5.     How to define and show charge transfer in the partition scheme based on DFTB? According to eq 7, CT is absorbed in electrostatic term.     

Reviewer 2 Report

To be published with minor corrections. Propose for publishing after minor spell check

Very nice paper. Recommend strongly for publication in the journal. Have not seen any gaps. Nevertheless, there are some misprints f.e.

somputing the screening (line 152)

also it would be better to define more rigorously meaning of segments and describe carefully the method of decomposing into the segments.

Round 2

Reviewer 1 Report

They have answered my questions properly.